# Insight into the Inhibitory Mechanisms of Hesperidin on α-Glucosidase through Kinetics, Fluorescence Quenching, and Molecular Docking Studies

**DOI:** 10.3390/foods12224142

**Published:** 2023-11-16

**Authors:** Kumaravel Kaliaperumal, Linyan Zhang, Liangliang Gao, Qin Xiong, Yan Liang, Yueming Jiang, Jun Zhang

**Affiliations:** 1National Engineering Research Center of Navel Orange, Gannan Normal University, Ganzhou 341003, China; kumarbio06@gmail.com (K.K.); 15899824665@139.com (L.Z.); togaoliangliang@outlook.com (L.G.); skyxiongqin@163.com (Q.X.); zjhxy110@126.com (Y.L.); ymjiang@scbg.ac.cn (Y.J.); 2Unit of Biomaterials Research, Department of Orthodontics, Saveetha Dental College, Saveetha Institute of Medical and Technical Sciences, SIMATS, Chennai 602105, India; 3South China Botanical Garden, Chinese Academy of Sciences, Guangzhou 510650, China

**Keywords:** hesperidin, α-glucosidase, antidiabetic, flavonoid, orange

## Abstract

The α-glucosidase inhibitor is of interest to researchers due to its association with type-II diabetes treatment by suppressing postprandial hyperglycemia. Hesperidin is a major flavonoid in orange fruit with diverse biological properties. This paper evaluates the effects of hesperidin on α-glucosidase through inhibitory kinetics, fluorescence quenching, and molecular docking methods for the first time. The inhibition kinetic analysis shows that hesperidin reversibly inhibited the α-glucosidase activity with an IC_50_ value of 18.52 μM and the inhibition was performed in an uncompetitive type. The fluorescence quenching studies indicate that the intrinsic fluorescence of α-glucosidase was quenched via a static quenching process and only one binding site was present between the hesperidin and α-glucosidase. The interaction between them was spontaneous and mainly driven by hydrogen bonds, as well as hydrophobic forces. Furthermore, the molecular docking results suggest that hesperidin might bond to the entrance or outlet part of the active site of α-glucosidase through a network of five hydrogen bonds formed between hesperidin and the four amino acid residues (Trp709, Arg422, Asn424, and Arg467) of α-glucosidase and the hydrophobic effects. These results provide new insight into the inhibitory mechanisms of hesperidin on α-glucosidase, supporting the potential application of a hesperidin-rich orange product as a hypoglycemic functional food.

## 1. Introduction

Diabetes mellitus (DM) is a metabolic illness presently influencing 425 million individuals around the world, and the overall number is expected to rise to 642 million by 2045 [1]. In addition, the DM disease is frequently accompanied by extreme complications, like cardiac problems, diabetic retinopathy, and neuropathy [2]. As the foremost common sort of DM, type II diabetes is recognized as having a close relationship with the occurrence of postprandial blood glucose [3]. In this manner, appropriate control of postprandial hyperglycemia is exceptionally essential for the control of type II diabetes.

α-glucosidase is an important catabolic enzyme that engages in the decomposition of oligosaccharides into monosaccharides. The glucose produced from starch by α-glucosidase-catalyzed decomposition is quickly assimilated into the blood, causing the occurrence of high levels of postprandial blood glucose [4]. Hence, α-glucosidase is specifically related to type II diabetes, and the retardation of the enzymatic activity of α-glucosidase might diminish the postprandial glucose levels, which could be an efficient treatment for type II diabetes [1,4]. Plenty of investigations have been carried out previously with the aim of discovering the efficient inhibitors of α-glucosidase to manage diabetes [5]. Although some synthetic medicines for type II diabetes treatment, which include metformin, glibenclamide, acarbose, and saxagliptin, are efficient to inhibit the enzymatic activity of α-glucosidase, they exert serious side effects, like hepatotoxicity and cardiac arrest [3,5]. Therefore, it is of much interest to search for an alternative antidiabetic modulator without any adverse effects on the health system. Some of the synthetic iminosugars with gauche side chain moieties were proved to be efficient in inhibiting the α-glucosidase enzyme [6]. Natural resources, such as traditional herbs and functional fruits, are abundant with biologically interesting compounds, which have been widely explored for the development of naturally occurring inhibitors on α-glucosidase. For example, narcissoside isolated from *Anoectochilus roxburghii* [7], trilobatin obtained from *Lithocarpus polystachyu* [8], prenylated flavonoids isolated from mulberry leaves [9], and furanolabdane diterpenoids from the tropical plant *Graptophyllum pictum* [10] are reported to exhibit an excellent inhibitory effect on α-glucosidase, showing high application potential as natural antidiabetic agents. 

Hesperidin, a flavonoid glycoside, is predominant in citrus fruits like orange, lemon, and grapefruit. Hesperidin is a natural plant defense molecule that protects the host plant against pests and microbial infections [11]. In addition, hesperidin has many biomedical properties, like anticancer, anticoagulant, antioxidant, and cardioprotection effects [12,13,14]. Hesperidin is found abundantly in both the juice and peel of orange fruits. For example, Gannan navel orange juice contains over 250 mg/L of hesperidin [15]. Orange peels are reported to have high hesperidin content, being about 42.56 mg per gram of dry weight of 95% ethanolic extract [16]. In most of the orange juice processing industries, the orange peel is discarded as a byproduct waste without recovering its hesperidin and other valuable products. Despite various biological effects that hesperidin possesses, the antidiabetic effect of hesperidin in lowering the blood glucose levels has still not been validated [17]. Previous studies have revealed that hesperidin exhibits a considerable inhibitory effect against α-glucosidase [18,19]; however, the inhibitory mechanism remains unknown, which greatly restricts its further application for the management of diabetes. With the rapid development of spectroscopic analysis and computer simulation techniques, multiple methods, such as the fluorescence spectrum, calorimetric analysis, and molecular modeling calculation, have been widely explored to elucidate the action mode between the small molecule and bio-macromolecule [8,9]. Therefore, in the present study, the inhibitory effect and the action mode of hesperidin binding to α-glucosidase were investigated through in vitro enzyme kinetics, a fluorescence quenching assay, and molecular docking studies for the first time. The results revealed here might shed light on the inhibitory mechanism of hesperidin on α-glucosidase, thereby greatly benefiting the application potential of hesperidin-rich orange products like orange juice and orange peel as natural hypoglycemic agents.

## 2. Materials and Methods

### 2.1. Chemicals and Instruments

Hesperidin was procured from Shanghai Yuanye Biotechnology Company (Yuanye, Shanghai, China). α-glucosidase (EC 3.2.1.20, from *Saccharomyces cerevisiae*) was obtained from Sigma-Aldrich Chemical. (St. Louis, MO, USA). Acarbose and *p*-nitrophenyl-α-D-glucopyranoside (*p*NPG) were purchased from Aladdin Industrial Corporation (Shanghai, China). De-ionized water was obtained from a Milli-Q Gradient A10 system (Millipore, Billerica, MA, USA). All the other chemicals and solvents used in the present study are of analytical grade from Damao Chemical Reagent Factory (Tianjin, China). A microplate reader (Tecan Spark 10M, Männedorf, Switzerland) was employed for the UV absorbance measurements. A fluorescence spectrophotometer (F-7000; Hitachi High-Tech Science Co., Tokyo, Japan) was used to quantify the fluorescence intensity.

### 2.2. In Vitro α-Glucosidase Inhibition Assay

An α-Glucosidase inhibition assay was carried out using a previous procedure with slight modifications [19]. In brief, to the wells of a 96-well microplate, α-glucosidase (50 μL, 0.185 U/mL) and phosphate buffer (40 μL, pH 6.8) were added, followed by the addition of hesperidin (20 μL) with various concentrations from 5 to 125 μg/mL. The reaction mixture was shaken well and incubated at 37 °C for 10 min. After that, 4 mM *p*NPG (50 μL) was added to the incubating reaction mixture as a substrate to trigger the enzymatic reaction, and a microplate reader (Tecan Spark 10M, Männedorf, Switzerland) was used to measure the absorbance of the mixture at 405 nm every 30 s for 20 min in total. Acarbose served as a positive control. Enzymatic activity without the presence of an inhibitor was defined as 100%. The relative enzymatic activity was calculated based on the following Equation (1).
(1)$$The\;relative\;enzymatic\;activity\;\left(\%\right)=\frac{R}{R_{0}}\times 100\%$$
where R and R_0_ are the slopes of the linear part of the reaction kinetics obtained with the presence or absence of hesperidin, respectively. The half inhibitory concentration (IC_50_) means the concentration of the samples causing a loss of 50% enzymatic activity. 

### 2.3. Kinetic Analysis of Enzyme Inhibition

The inhibition kinetic analysis was performed by the same procedure as that described for the in vitro α-glucosidase inhibition assay. For the reversibility test, the concentration of substrate *p*NPG was kept at 4 mM, while the hesperidin concentration was set to 0, 6.14, 12.28, 1.38, and 20.47 × 10^−5^ M, respectively. The plot of the velocity of the enzymatic reaction versus the α-glucosidase concentration with the presence of different concentrations of hesperidin was constructed to determine the reversibility of the inhibitor. For the inhibition type test, the α-glucosidase concentration was kept at 0.96 μM, while the concentration of hesperidin was set to 0, 6.14, 12.28, 1.38, and 20.47 × 10^−5^ M, respectively. Lineweaver–Burk plots were then constructed by the reciprocal velocity. the reciprocal substrate concentration with the presence of different concentrations of hesperidin. The values of the Michaelis–Menten constant (*K*_m_) and the maximum velocity (*V*_max_) were determined by the Lineweaver–Burk plots according to Equation (2) [20]. The values of the new kinetic constants *K*_ik_ and *K*_iv_, which reflected the changes of *K*_m_ and *V*_max_ in response to the inhibitor concentrations, were obtained by Equations (3) and (4), respectively, based on Yang’s method [21].
(2)$$\frac{1}{V}=\frac{K_{m}}{V_{max}}\;\frac{1}{\left[S\right]}+\frac{1}{V_{max}}$$
(3)$$\frac{1}{K_{m}}=\frac{1}{K_{m,0}}\;\times \left(1+\frac{\left[I\right]}{K_{ik}}\right)$$
(4)$$\frac{1}{V_{max}}=\frac{1}{V_{max,0}}\;\times \left(1+\frac{\left[I\right]}{K_{iv}}\right)$$
where *K*_m,0_ and *V*_max,0_ are the Michaelis–Menten constant and the maximum catalytic velocity of the enzyme without the presence of hesperidin, respectively. [*S*] and [*I*] represent the concentrations of the substrate *p*NPG and the inhibitor hesperidin, respectively.

### 2.4. Fluorescence Spectroscopic Analysis

A fluorescence quenching experiment was used to investigate the interaction between the α-glucosidase enzyme and hesperidin based on the methodology of He et al. [8]. Different amounts of stock solution (25 mM in dimethyl sulfoxide) of hesperidin were titrated into a 3 mL α-glucosidase enzyme solution (0.96 μM) to achieve a series of final concentrations of 0.55, 1.09, 2.19, 3.28, 4.37, and 5.46 × 10^−5^ M for hesperidin, respectively. The corresponding enzyme solution without the addition of hesperidin was used as a blank control. At each concentration after the titration of hesperidin, the intrinsic fluorescence of the α-glucosidase enzyme was scanned using a fluorescence spectrophotometer at a wavelength of 300–500 nm after being excited at 280 nm. The emission and excitation slit width was set to 2.5 nm. The fluorescence spectra were recorded at three different temperatures (298, 304, and 310 K). All the fluorescence data were corrected based on Equation (5) to subtract the internal filtering effect caused by the ultraviolet absorption [8].
(5)$$F_{corr}=F_{obs}e^{\frac{{(A}_{1}+A_{2})}{2}}$$
where *F*_obs_ and *F*_corr_ represent the observed and corrected fluorescence intensities, respectively. *A*_1_ and *A*_2_ represent the absorbance of the corresponding solution at excitation and the emission wavelength, respectively.

### 2.5. Molecular Docking

The molecular model of α-glucosidase (PDB ID: 7kad) was obtained from Protein Data Ban (https://www1.rcsb.org/, accessed on 6 August 2023). The 3D structure of hesperidin (compound CID: 10621) was derived from PubChem (https://pubchem.ncbi.nlm.nih.gov/, accessed on 6 August 2023) Database. The obtained molecular model of α-glucosidase was preprocessed using Autodock Tools 1.5.7 software to remove water, add hydrogen atoms, and perform charge calculation. The molecular docking calculation was performed using Auto Dock vina 1.2.3 software. The exhaustiveness parameter was set to 16 during the docking process while other parameters were kept as default values of the software. The binding mode with the lowest energy was selected as the most favorable conformation for further analysis. The molecular docking results were visualized and analyzed using both PyMol 2.6 and LigPlot software (version 2.2) [22].

### 2.6. Statistical Analysis

All the experimental results were collected based on triplicate assays, and the mean ± standard deviation errors were recorded. A one-way analysis of variance (ANOVA) was employed to analyze the data using the SPSS (Version 6.0) statistical package at *p* < 0.05.

## 3. Results and Discussions

### 3.1. In Vitro Antidiabetic Effect of Hesperidin on α-Glucosidase Enzyme

Hesperidin is a flavonoid glycoside containing one glucose and one rhamnose as subunits (Figure 1A) and is abundantly present in orange fruits [15]. In the present study, the inhibitory effect of hesperidin on α-glucosidase was assessed biochemically. As shown in Figure 1B, the enzymatic activity of α-glucosidase was significantly inhibited by hesperidin, and the relative activity was gradually reduced, along with the increase of the hesperidin concentration, showing a dose-dependent manner. The enzymatic activity was almost completely inhibited when the hesperidin concentration was over 200 μM. The IC_50_ value of hesperidin on enzyme inhibition was found to be 18.52 ± 1.26 µM based on the dose–activity curve, which is in line with a previous report showing an IC_50_ of 15.75 μg/mL (25.8 μM) [19]. Interestingly, the inhibitory effect of hesperidin was comparable to the positive control acarbose (IC_50_ 12.24 μM), indicative of the great application potential of hesperidin as a substituent for acarbose. 

Flavonoids are natural compounds widely present in plants and they possess immense biological value in terms of antioxidant, anticancer, and antidiabetic properties [23,24]. Citrus plant-contained flavonoids like hesperidin, naringin, and nobiletin are proven to be effective against many metabolic disorders, like hypertension and cardiovascular disease [25]. In an earlier finding, it was observed that hesperidin exhibited a selective and significant α-glucosidase inhibitory effect and also reduced the glucose-6-phosphate enzyme activity in HepG2 cells [26]. It has also been proven that drinking orange juice consecutively reduces the postprandial blood glucose level due to the hesperidin content of orange juice [27].

### 3.2. Kinetic Analysis of Enzyme Inhibition

The reversibility of the inhibitory effect was investigated by the plot of the α-glucosidase concentration ([α-glucosidase]) versus the velocity (V) of the enzymatic reaction. As shown in Figure 2A, the velocity gradually increased along with the increase in the enzyme concentration at the same concentration of hesperidin. Meanwhile, all the lines pass through the origin with good linearity, and the slope decreases with the increase of the hesperidin concentration. These results indicate that α-glucosidase was not completely inactivated, but the catalytic rate of the enzyme was decreased by hesperidin, suggesting that hesperidin was a reversible inhibitor against the α-glucosidase enzyme, according to previous studies [20,28]. It was reported that irreversible inhibitors completely inactivate enzymes by forming stable complexes via covalent intermolecular interactions [29]. Furthermore, the Lineweaver–Burk double-reciprocal plot was used to determine the inhibition type of hesperidin on α-glucosidase. As shown in Figure 2B, a family of parallel lines was obtained by plotting 1/V against 1/[*p*NPG], suggestive of the type of uncompetitive inhibition [20]. 

Meanwhile, as shown in Table 1, both the *K*_m_ and *V*_max_ values acquired from Equation (2) gradually decreased with the increase of the hesperidin concentration, which is in line with the uncompetitive inhibition kinetics of cinnamic acid amide on α-glucosidase [30]. To further confirm the uncompetitive inhibition type of hesperidin on α-glucosidase, two new kinetic constants, *K*_ik_ and *K*_iv_, were calculated from Equations (3) and (4). According to Yang’s method, a ratio of *K*_iv_/*K*_ik_ below 2.0 suggests that the inhibition is an uncompetitive type, over 5.0 indicates noncompetitive or competitive, and from 2.0 to 5.0 implies a mixed-type inhibition [21]. As shown in Table 1, the ratio of *K*_iv_/*K*_ik_ was around 1.0 at different concentrations of hesperidin, which is indicative of an uncompetitive inhibition. Natural products as uncompetitive inhibitors of α-glucosidase have not been reported much so far [5]. The only a few examples reported previously include vitexin [31], corosolic acid [20], cinnamic acid amide [30], and ginsenoside Rg1 [32]. The present discovery of hesperidin as an excellent uncompetitive inhibitor is very interesting since this type of inhibitor was generally more efficient than the competitive and noncompetitive inhibitors in the in vivo assay [33].

### 3.3. Fluorescence Quenching Assay

The intrinsic fluorescence of α-glucosidase is mainly attributable to the presence of Trp (Tryptophan) and Tyr (tyrosine) residues, which emit fluorescence at an excitation wavelength of 280 nm [20]. Figure 3A clearly depicts that in the presence of α-glucosidase only, the highest emission peak was at 345 nm when the excitation wavelength was 280 nm, which is in good agreement with previous reports [31,32]. Hesperidin had no interference on the α-glucosidase fluorescence because it did not emit fluorescence under the same conditions (curve h of Figure 3A). The fluorescence intensity of α-glucosidase gradually decreased with the increase of the hesperidin concentration (curves a–g, Figure 3A), suggesting that the conformation of the α-glucosidase might be modified due to the interaction between them. Therefore, the quenching mechanism was further investigated using the Stern–Volmer equation (Equation (6)).
(6)$$\frac{F_{0}}{F}=1+K_{sv}[Q]=1+K_{q}\tau_{0}[Q]$$

In the formula above, *F* and *F*_0_ represent the corrected fluorescence intensity of α-glucosidase in the presence and absence of the inhibitor, respectively; [*Q*] is the concentration of the inhibitor; *K*_q_ and *K*_SV_ represent the bimolecular quenching rate constant and quenching constant, respectively; and *τ*_0_ represents the fluorophore lifetime of a free biomacromolecule, which is approximately 10^−8^ s for α-glucosidase [34].

As shown in Figure 3B, the plot of *F*_0_/*F* versus [*Q*] exhibits good linearity at all three temperatures (298, 304, and 310 K), indicating that the quenching process by hesperidin was a single static or dynamic quenching [35]. The slope of the straight line represents the *K*_SV_ value at the corresponding temperature, and the *K*_SV_ values at all three temperatures are listed in Table 2. The *K*_SV_ value decreased with the increase in temperature, which is a characteristic of the static quenching process [36]. In addition, all the corresponding *K*_q_ values were of the 10^12^ magnitude order, which is much larger than the maximum diffusion collision quenching rate constant of the biological macromolecules (2.0 × 10^10^ L mol^−1^ s^−1^), further confirming that the quenching process of α-glucosidase by hesperidin was a static quenching [31,37]. The present result suggests the fluorescence quenching by hesperidin should be achieved via the formation of a ground-state complex rather than a dynamic collision process.

By using the fluorescence quenching data, the binding constant (*K*_a_) and the number (*n*) of the binding site were obtained. Equation (7) [38] was also used to determine that [*P*_t_] and [*Q*_t_] were the concentrations of α-glucosidase and hesperidin, respectively.
(7)$$log\frac{F_{0}-F}{F}=nlogK_{a}-nlog\;\frac{1}{\left[Q_{t}\right]-\frac{\left(F_{0\;}-F\right)[P_{t}]}{F_{0}}}$$

According to Equation (7), the values of *K*_a_ at 298, 304, and 310 K were calculated. As shown in Table 2, all the *K*_a_ values had a magnitude order of 10^4^ L mol^−1^, suggesting a strong binding affinity between α-glucosidase and hesperidin [31]. In addition, it can be seen from the data in Table 2 that the *K*_a_ value decreased with the increase in the reaction temperature, indicating that the stability of the inhibitor–enzyme complex decreased with the increase in temperature, and this result further confirms that the fluorescence quenching by hesperidin was a static process. Furthermore, the *n* values (1.07, 1.06, and 0.96) obtained at three different temperatures were all close to 1 (Table 2), suggesting that there was only one binding site between hesperidin and α-glucosidase [20,39]. 

To further identify the driving forces for the formation of the α-glucosidase–hesperidin complex, the thermodynamic parameters involving enthalpy change (Δ*H°*), entropy change (Δ*S°*), and the Gibbs free energy (Δ*G°*) of the binding process were calculated according to the Equations (8) and (9) (van’t Hoff equation), where *R* represents the gas constant (8.314 J mol^−1^ K^−1^), and *K*_a_ is the binding constant at the corresponding temperature.
(8)$$logK_{a}=-\frac{∆H{}^{\circ}}{2.303RT}+\frac{∆S{}^{\circ}}{2.303RT}$$
(9)∆G°=∆H−T∆S°

In the present study, since the variation range of the temperature was narrow, the enthalpy change (Δ*H°*) can be considered as a constant [7]. The main non-covalent interactions between the enzymes and inhibitors include electrostatic interaction, hydrophobic forces, hydrogen bonds, and van der Waals forces. As shown in Table 2, the Δ*G°* of the present study was around −25 KJ mol^−1^ at all three temperatures, less than zero (Δ*G°* < 0), indicating that the binding of hesperidin with α-glucosidase was a spontaneous procedure [29]. Furthermore, Δ*H°* < 0 and Δ*S°* > 0 (Table 2) suggest that the dominant driving forces for the binding procedure of hesperidin with α-glucosidase were hydrogen bonds and hydrophobic interactions [31,40].

### 3.4. Molecular Docking

Molecular docking was used to further determine the interaction between α-glucosidase and hesperidin (Figure 4). The lowest binding energies of the α-glucosidase–hesperidin complex was −11.375 kcal/mol, indicating that the interaction between α-glucosidase and hesperidin was strong and occurred spontaneously, which is in line with the result of the thermodynamic investigation showing Δ*G°* < 0. As shown in Figure 4B, hesperidin formed five hydrogen bonds with the amino acid residues involving Trp709, Arg422, Asn424, and Arg467, with the distance ranging from 2.3 to 2.9 Å. Moreover, hesperidin was surrounded by a serial of amino acid residues, such as Asp430, Arg705, Glu707, Arg426, Arg467, Trp423, and Asn424, via hydrophobic forces (Figure 4C). These observations indicate that hydrogen bonds and hydrophobic forces were the main driving forces between hesperidin and α-glucosidase, which is in good agreement with the results of the thermodynamic analysis. Many previous studies have reported that amino acid residues involving Tyr 158, Phe 159, Asp 215, Glu 277, Arg 312, Asp 352, and Glu 411 play critical roles in binding the substrate, and they might form an enzymatically active site of α-glucosidase [41,42]. The present docking study indicates that hesperidin bound to sites other than the active center of α-glucosidase, which is consistent with the results obtained in the kinetic analysis showing that hesperidin was an uncompetitive inhibitor. 

As an uncompetitive inhibitor, hesperidin only binds with the enzyme–substrate complexes, showing no competition to the active center of the enzyme. Therefore, the present results allow us to propose that the hesperidin might bond to the entrance or outlet part of the active center, thereby obstructing the release of the substrate and catalytic reaction product from the active center, eventually inhibiting the enzymatic activity of α-glucosidase. Similarly, uncompetitive inhibitors, like cinnamic acid amide [30], corosolic acid [20], and ginsenoside Rg1 [32], are suggested to bond to the entrance of the active center of α-glucosidase based on the results of the molecular docking studies. However, more approaches, such as three-dimensional fluorescence spectrum, a time-resolved fluorescence technique, and CD (circular dichroism) spectrum, should be employed in future to further identify the conformational change of the α-glucosidase enzyme molecule after the interaction with hesperidin. 

## 4. Conclusions

The present research elucidates the inhibitory mechanism of hesperidin on α-glucosidase through enzyme kinetics, thermodynamics, and a molecular modeling approach for the first time. Hesperidin was found to exhibit a significant inhibitory effect on the enzymatic activity of α-glucosidase through a reversible and uncompetitive inhibition mode. The fluorescence quenching results indicate that hesperidin spontaneously bonded to α-glucosidase with one binding site by non-covalent interactions, which mainly consisted of hydrogen bonds and hydrophobic effects. The molecular docking studies suggest that hesperidin might bond to the entrance or outlet part of the active center of the enzyme, thus blocking the release of the substrate and catalytic product. Further research via in vivo experiments involving the efficacy and safety evaluations would be much more supportive of the antidiabetic application of hesperidin. Meanwhile, some of the other major flavonoids present in orange fruit, such as narirutin, didymin, sinensetin, and nobiletin, should be assessed for their synergistic effects alongside hesperidin for inhibiting the α-glucosidase effect.

## Figures and Tables

**Figure 1 foods-12-04142-f001:**
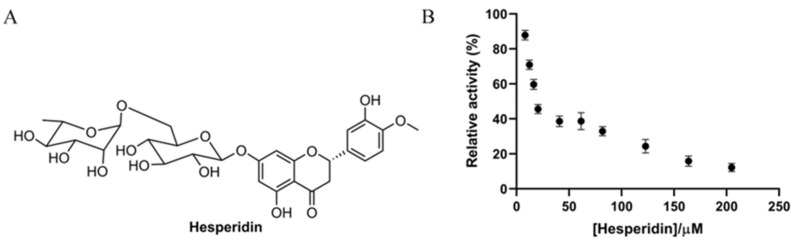
(**A**) Chemical structure of hesperidin. (**B**) Effect of different concentrations of hesperidin on the relative activity of α-glucosidase.

**Figure 2 foods-12-04142-f002:**
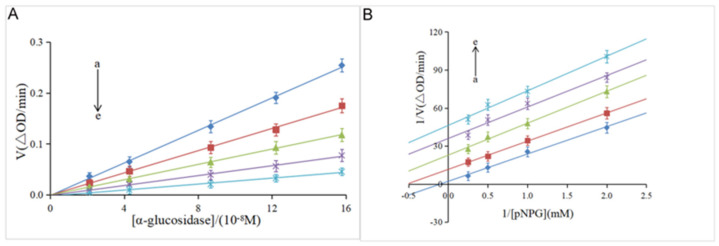
(**A**) Plots of V versus α-glucosidase. *p*NPG = 4 mM; hesperidin = 0, 6.14, 12.28, 16.38, 20.47 × 10^−5^ M for curves a→e (from light sky blue to light sea green), respectively. (**B**) Lineweaver–Burk plots. α-glucosidase = 0.96 μM; hesperidin = 0, 6.14, 12.28, 16.38, 20.47 × 10^−5^ M for curves a→e (from light sea green to light sky blue), respectively.

**Figure 3 foods-12-04142-f003:**
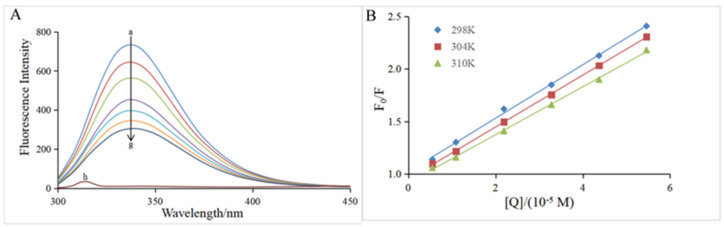
(**A**) Fluorescence spectra of α-glucosidase at varying concentrations of hesperidin. *T* = 298 K; λ_ex_ = 280 nm; α-glucosidase = 0.96 μM; hesperidin = 0, 0.55, 1.09, 2.19, 3.28, 4.37, 5.46 × 10^−5^ M for curves a→g, respectively. Curve h displays the emission spectrum of hesperidin at the concentration of 0.55 × 10^−5^ M. (**B**) The Stern–Volmer plots at three different temperatures (*T* = 298, 304, 310 K).

**Figure 4 foods-12-04142-f004:**
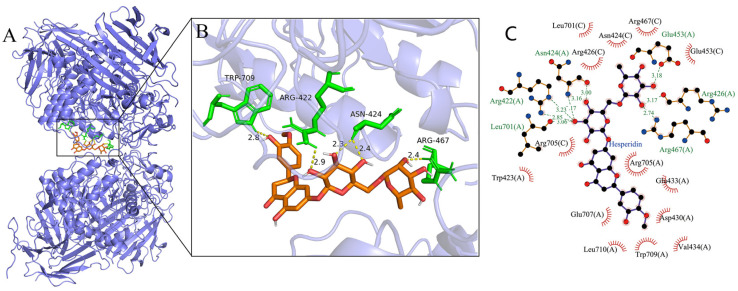
(**A**) Molecular simulation diagram of hesperidin–α-glucosidase complex. (**B**) The hydrogen bonds between hesperidin and α-glucosidase. (**C**) The hydrophobic interactions formed between hesperidin and α-glucosidase.

**Table 1 foods-12-04142-t001:** Effect of different concentrations of hesperidin on *V*max, *K*m of α-glucosidase, and ratios of the *K*iv to *K*ik using *p*NPG (*p*-nitrophenyl-α-D-glucopyranoside) as substrate.

[*I*](Hesperidin)/µM	*V* _max_	*K* _m_	*K*_iv_/*K*_ik_
0	0.29	6.70	//
61.42	0.10	2.44	0.92
122.84	0.04	0.91	1.02
163.78	0.03	0.90	0.96
201.73	0.02	0.46	1.01

**Table 2 foods-12-04142-t002:** Effect of temperatures on the fluorescence quenching constant (*K*_SV_), binding constant (*K*_a_), binding site number (*n*), and the thermodynamic parameters of hesperidin α-glucosidase complex.

*T* (K)	*K*_SV_ (×10^5^ L mol^−1^)	*K*_a_ (×10^4^ L mol^−1^)	*n*	Δ*H*0 (kJ mol^−1^)	Δ*G*0 (kJ mol^−1^)	Δ*S*0 (J mol^−1^ K^−1^)
298	2.58 ± 0.01 ^a^	2.97 ± 0.04 ^a^	1.07 ± 0.05 ^a^	−22.82 ± 2.06	−25.51 ± 0.03 ^a^	9.03 ± 0.08
304	2.40 ± 0.05 ^b^	2.44 ± 0.25 ^b^	1.06 ± 0.01 ^a^		−25.56 ± 0.02 ^a^	
310	2.11 ± 0.18 ^c^	2.08 ± 0.17 ^c^	0.96 ± 0.04 ^a^		−25.62 ± 0.02 ^a^	

Data are presented as mean ± SD (*n* = 3). Different lowercase letters following the data within the same column indicate significant statistical difference (*p* < 0.05).

## Data Availability

The data presented in this study are available on request from the corresponding author.

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
