# Peer review of "Insight into the Inhibitory Mechanisms of Hesperidin on α-Glucosidase through Kinetics, Fluorescence Quenching, and Molecular Docking Studies"

_foods, 2023, doi:10.3390/foods12224142_

Round 1
Reviewer 1 Report
Comments and Suggestions for Authors
The manuscript entitled ‘An Insight into the Inhibitory Mechanisms of Hesperidin on α-Glucosidase through Kinetics, Fluorescence Quenching and Molecular Docking Studies’ is interesting and well written. Inhibition of this important carbohydrate digestive enzyme can help to reduce blood glucose levels and remedy the onset of Type 2 diabetes. Natural glucosidase inhibitors are gaining much interest. Hesperidin is a flavonoid glycoside that also possesses antioxidant effects and can equally interfere in oxidative stress reduction. The authors have evaluated kinetic studies and molecular docking which gives an insight of structure-activity relationship. The test compound is a component of many fruits and food plants and diabetes is a metabolic illness.
1. Line 36…correct to ….. the number is expected to rise to 642 million by 2045
2. Line 38…kidney failure not disappointment
3. Line 177…Hesperidin is a flavonoid glycoside containing…
4. Line 358…supporting the potential application of orange fruits……
Find some recommended and citable literature for the enrichment of your introduction and discussion.
Molecules 2023, 28(12), 4802; https://doi.org/10.3390/molecules28124802
Author Response
Comment 1: The manuscript entitled ‘An Insight into the Inhibitory Mechanisms of Hesperidin on α-Glucosidase through Kinetics, Fluorescence Quenching and Molecular Docking Studies’ is interesting and well written. Inhibition of this important carbohydrate digestive enzyme can help to reduce blood glucose levels and remedy the onset of Type 2 diabetes. Natural glucosidase inhibitors are gaining much interest. Hesperidin is a flavonoid glycoside that also possesses antioxidant effects and can equally interfere in oxidative stress reduction. The authors have evaluated kinetic studies and molecular docking which gives an insight of structure-activity relationship. The test compound is a component of many fruits and food plants and diabetes is a metabolic illness.
Response: Thanks for your kind encouragements.
Comment 2: Line 36…correct to ….. the number is expected to rise to 642 million by 2045.
Response: Thanks. As per review comment, the sentence was modified in the revised manuscript.
Comment 3: Line 38…kidney failure not disappointment..
Response: Thanks for the comments. As per review comments, this sentence was rewritten to correct grammar errors.
Comment 4: Line 177…Hesperidin is a flavonoid glycoside containing…
Response: Thanks. As per review comments, this grammar error was corrected in the revised manuscirpt.
Comment 5: Line 358…supporting the potential application of orange fruits…….
Response: Thanks. The authors have corrected this error based on your comment.
Comment 6: Find some recommended and citable literature for the enrichment of your introduction and discussion. Molecules 2023, 28(12), 4802; https://doi.org/10.3390/molecules28124802.
Response: Thanks. As per review comments, the authors have added the recommended article in the introduction section, and corresponding modifications were made as well in the Reference section.
Reviewer 2 Report
Comments and Suggestions for Authors
Hesperidin is a natural compound found in orange fruits that has various health benefits. This article investigates how hesperidin can inhibit the activity of α-glucosidase, an enzyme that breaks down carbohydrates and increases blood sugar levels. The authors used different methods to study the interaction between hesperidin and α-glucosidase, such as measuring the inhibition rate, observing the fluorescence change, and simulating the molecular docking. They found that hesperidin can bind to the enzyme and reduce its activity in a reversible and uncompetitive way. They also found that hesperidin can interact with the enzyme through hydrogen bonds and hydrophobic forces, and that it can block the entrance or exit of the active site of the enzyme. The authors concluded that hesperidin can be a potential anti-hyperglycemic agent that can prevent or treat type-II diabetes by lowering blood sugar levels after meals. They suggested that orange fruits rich in hesperidin can be used as functional foods for diabetic patients. They also recommended further research on the effects of hesperidin in vivo and on other natural compounds that may have similar or synergistic effects on α-glucosidase.
Here are the suggestions which should be answered:
1. The abstract is well-written and summarizes the main findings and implications of your study. However, you may want to avoid repeating the same sentences in the last paragraph of the abstract and the conclusion section. You could either shorten the abstract or expand the conclusion with some additional points or suggestions for future research.
- The introduction provides a clear background and motivation for your study. You have cited relevant literature and explained the objectives and hypotheses of your research. However, you may want to state your research questions more explicitly and briefly mention the methods and results that you used to answer them.
- It is suggested to mention the importance of synthetic efforts towards development of antidiabetic medicine. In this regard, it is recommended to emphasis the importance of iminosugars and sugar derivatives as an antidiabetic agent and suggested to cite relevant articles in the introduction section; eg: https://doi.org/10.1002/anie.202217809
- The discussion section interprets and discusses the results of your study in relation to your research questions and hypotheses. You have compared your findings with those of previous studies and explained the possible mechanisms and implications of your results. However, you may want to address some of the limitations and challenges of your study, such as the validity and generalizability of your results, the sources of error and uncertainty in your measurements, the assumptions, and parameters that you used in your models, etc.
- The conclusion section summarizes the main findings and contributions of your study. You have also suggested some potential applications of hesperidin-rich orange products as hypoglycemic functional foods. However, you may want to provide some specific recommendations or directions for future research on this topic, such as testing the efficacy and safety of hesperidin in vivo, exploring other flavonoids or natural compounds that may have similar or synergistic effects on α-glucosidase, investigating the molecular interactions between hesperidin and other enzymes or receptors involved in glucose metabolism, etc.
- How did you determine the type of inhibition kinetics of hesperidin? Did you use any mathematical model or software to fit the data?
- How did you calculate the thermodynamic parameters of the interaction between hesperidin and α-glucosidase? What equations and assumptions did you use?
- How did you compare the inhibitory effects of hesperidin with other known α-glucosidase inhibitors? Did you use any statistical tests or criteria to evaluate the significance of the differences?
Author Response
Comment 1: The abstract is well-written and summarizes the main findings and implications of your study. However, you may want to avoid repeating the same sentences in the last paragraph of the abstract and the conclusion section. You could either shorten the abstract or expand the conclusion with some additional points or suggestions for future research.
Response: Thanks. As per review comments, the authors have deleted the repeated expressions in the conclusion section, and added some sentences describing the future research prospect. It reads ‘Meanwhile, some of other major flavonoids present in orange fruit such as narirutin, didymin, sinensetin, and nobiletin should be assessed alongside hesperidin for inhibiting the a-glucosidase effect in terms of their synergistic effects.’.
Comment 2: The introduction provides a clear background and motivation for your study. You have cited relevant literature and explained the objectives and hypotheses of your research. However, you may want to state your research questions more explicitly and briefly mention the methods and results that you used to answer them.
Response: Thanks. Based on your suggestions, the authors have added the following sentence in the revised manuscript to briefly mention the methods used for investigating the inhibitory mechanisms. It reads ‘With the rapid development of spectroscopic analysis and computer simulation techniques, multiple methods such as the fluorescence spectrum, calorimetric analysis, and molecular modeling calculation, have been widely explored to elucidate the action mode between small molecule and bio-macromolecule.’
Comment 3: It is suggested to mention the importance of synthetic efforts towards development of antidiabetic medicine. In this regard, it is recommended to emphasis the importance of iminosugars and sugar derivatives as an antidiabetic agent and suggested to cite relevant articles in the introduction section; eg: https://doi.org/10.1002/anie.202217809.
Response: Thanks for the comments. As per review comments the synthetic drugs as antidiabetic medicine and the recommended literature were added in the revised manuscript. It reads ‘Some of the synthetic iminosugars with gauche side chain moieties were proved to be efficient in inhibiting the a-glucosidase enzyme [6].’
Comment 4: The discussion section interprets and discusses the results of your study in relation to your research questions and hypotheses. You have compared your findings with those of previous studies and explained the possible mechanisms and implications of your results. However, you may want to address some of the limitations and challenges of your study, such as the validity and generalizability of your results, the sources of error and uncertainty in your measurements, the assumptions, and parameters that you used in your models, etc.
Response: Thanks. According to your suggestion, the authors have added some sentences indicating the limitations of the techniques used in the present study. It reads ‘However, more approaches such as three-dimensional fluorescence spectrum, time-resolved fluorescence technique, and CD (circular dichroism) spectrum, should be employed in future to further identify the conformational change of the a-glucosidase enzyme molecule after the interaction with hesperidin.’ .
Comment 5: The conclusion section summarizes the main findings and contributions of your study. You have also suggested some potential applications of hesperidin-rich orange products as hypoglycemic functional foods. However, you may want to provide some specific recommendations or directions for future research on this topic, such as testing the efficacy and safety of hesperidin in vivo, exploring other flavonoids or natural compounds that may have similar or synergistic effects on α-glucosidase, investigating the molecular interactions between hesperidin and other enzymes or receptors involved in glucose metabolism, etc.
Response: Thanks for the comments. As per review comments the conclusion part is given more about the other flavonoids in citrus as antidiabetic agent and the potent synergetic effects. It reads ‘Further research on in vivo experiments involving the efficacy and safety evaluations would be much more supportive of the antidiabetic application of hesperidin. Meanwhile, some of other major flavonoids present in orange fruit such as narirutin, didymin, sinensetin, and nobiletin should be assessed alongside hesperidin for inhibiting the a-glucosidase effect in terms of their synergistic effects.’
Comment 6: How did you determine the type of inhibition kinetics of hesperidin? Did you use any mathematical model or software to fit the data?.
Response: Thanks for the question. The reversibility of the inhibitory effect was investigated by the plot of α-glucosidase concentration ([α-glucosidase]) versus the velocity (V) of the enzymatic reaction (Figure 2A). Furthermore, the Line-weaver-Burk double-reciprocal plot was used to determine the inhibition type of hesperidin on α-glucosidase (Figure 2B). The data were processed by using the SPSS (Version 6.0) statistical package and analyzed based on the Equations 2-4 showing in the manuscript.
Comment 7: How did you calculate the thermodynamic parameters of the interaction between hesperidin and α-glucosidase? What equations and assumptions did you use?.
Response: Thanks for the question. The thermodynamic parameters of the interactions between hesperidin and α-glucosidase is determined by the fluorescence quenching assay and the mathematical equations involving Eqs. 5-9 (as shown in manucscript) are used to analyze the data obtained.
Comment 8: How did you compare the inhibitory effects of hesperidin with other known α-glucosidase inhibitors? Did you use any statistical tests or criteria to evaluate the significance of the differences?.
Response: Thanks for the question. The inhibition effects of hesperidin were compared with acarbose. The inhibitory effect of hesperidin (IC50 18.52 µM) was comparable to the positive control acarbose (IC50 12.24 mM). The data obtained in the present study were analyzed by one-way analysis of variance (ANOVA) using the SPSS (Version 6.0) statistical package at P < 0.05.
Reviewer 3 Report
Comments and Suggestions for Authors
The manuscript: "An Insight into the Inhibitory Mechanisms of Hesperidin on α-Glucosidase through Kinetics, Fluorescence Quenching and Molecular Docking Studies" represents a solid study for evaluation of inhibitory mechanism of hesperidin on α-glucosidase by using various approaches. Also, this study could have further application for revealing hesperidin-enriched plants with anti-diabetic potential.
The following minor changes should be made:
Introduction
Pg 1, Line 37-38: Starting a sentence with abbreviation should be avoided, as in the following: "DM is frequently accompanied by extreme complications, including cardiovascular infections, neuropathy, retinopathy, and kidney disappointment [2]."
Pg 2, Line 45-48: Provide appropriate citation for the following: "Hence, α-glucosidase is specifically related to type II diabetes, and the retardation of the enzymatic activity of α-glucosidase might diminish the postprandial glucose levels, which could be an efficient treatment for type II diabetes."
Conclusion
Conclusion should be revised, since it represents rephrased version of the Abstract. This section should represent key findings and conclusion of the study.
Author Response
Comment 1: Pg 1, Line 37-38: Starting a sentence with abbreviation should be avoided, as in the following: "DM is frequently accompanied by extreme complications, including cardiovascular infections, neuropathy, retinopathy, and kidney disappointment [2]."
Response: Thanks. As per review comment, the authors have modified this sentence in the revised manuscrpt. It reads ‘In addition, the DM disease is frequently…….’.
Comment 2: Line 45-48: Provide appropriate citation for the following: "Hence, α-glucosidase is specifically related to type II diabetes, and the retardation of the enzymatic activity of α-glucosidase might diminish the postprandial glucose levels, which could be an efficient treatment for type II diabetes."
Response: Thanks. Based on your suggestion, the authors have added related references in the revised manuscript.
Comment 3: Conclusion should be revised, since it represents rephrased version of the Abstract. This section should represent key findings and conclusion of the study.
Response: Thanks. As per review comments the conclusion part was revised and the repeated descriptions was deleted.